# Application of iTRAQ Technology to Identify Differentially Expressed Proteins of Sauce Lamb Tripe with Different Secondary Pasteurization Treatments

**DOI:** 10.3390/foods11081166

**Published:** 2022-04-18

**Authors:** Ning An, Ran Hou, Yangming Liu, Ping Han, Wei Zhao, Wenxia Wu, Shiling Lu, Hua Ji, Juan Dong

**Affiliations:** 1College of Food Quality and Safety, Shihezi University, Shihezi 832000, China; an1727036614@163.com (N.A.); 18899592269@163.com (R.H.); hpshzdx@163.com (P.H.); wwx2512467283@163.com (W.W.); lushiling_76@163.com (S.L.); jh_food@shzu.edu.cn (H.J.); 2Sesame Research Center, Henan Academy of Agricultural Sciences, Zhengzhou 450002, China; pl20200042@163.com; 3School of Food Science and Technology, Jiangnan University, Wuxi 214122, China; zhaow@jiangnan.edu.cn

**Keywords:** heat treatment (HT), high-pressure processing (HPP), iTRAQ technology, sauce lamb tripe, texture

## Abstract

Vacuum-packed sauce lamb tripe was subjected to secondary pasteurization by high-pressure processing (HPP) and heat treatment (HT), and iTRAQ technology was applied to investigate the differentially expressed proteins (DEPs). The analysis revealed 484 and 398 DEPs in the HPP and HT samples, respectively, compared with no treatment. These DEPs were sorted by texture results, and it was revealed that these DEPs acted in different biological processes with many structural proteins and protein subunits related to lamb tripe texture. The results verified by Western blot were consistent with the protein expression changes observed by proteomics. The bioinformatics analysis showed that the hardness and gumminess of the sauce lamb tripe after HT might be related to changes in the expression of CNN1 and FN1. The changes in the expression of TMP, FN1, YWHAG, TTN, collagen isoforms, and ARPC3 might be related to the improved springiness and chewiness of lamb tripe after HPP.

## 1. Introduction

Meat prepared in soy sauce is widely consumed in southern China due to its unique flavor and attractive color [1]. In recent years, many different types of Chinese sauce products have been developed, and sauce lamb tripe is one of the most popular products. The muscle of lamb tripe is mainly smooth, which has a good taste. Marinating the tripe in soy sauce provides additional flavor and color. Sauce lamb tripe is relatively rich in nutritional value; therefore, it is more susceptible to microbial spoilage. Thus, vacuum packaging and secondary sterilization can be used to increase the storage time and improve the quality of sauce lamb tripe.

High-pressure processing (HPP), a non-thermal processing technology, is frequently used to extend product shelf life and improve quality [2]. HPP can change the content of amino acids, lactic acid, and other metabolites in soy sauce meat [3], and can also affect the fat oxidation of the meat by affecting lipase activity [4]. The effects on metabolites and fat oxidation can correspond to changes in amino acids or enzymes, allowing protein analysis to detect the changes in the quality of the sauce products treated by HPP.

Proteomics is a powerful emerging technology that can be applied to meat products to reveal the underlying biochemical mechanisms of quality change [5]. Proteomics can identify the differences in proteins between meats of different quality, and meats subjected to different processing methods. Early proteomic studies used two-dimensional gel electrophoresis, but it is difficult to apply this method to identify proteins that are too large or small, too acidic or basic, extremely hydrophobic, or in low abundance [6]. As an alternative, iTRAQ technology has become an important tool for proteomics research. Shi, Zhang, Lei, Shen, Yu and Luo (2018) [7] used iTRAQ technology to study the correlation between protein differences and the quality of shrimp subjected to different treatments. They found 12 kinds of proteins related to the quality of shrimp; these were associated with protein structure, metabolic enzymes, and protein turnover. Wei, Li, Zhang and Liu (2019) [8] identified protein differences between beef samples of different quality through iTRAQ technologies, and found that changes in meat quality correlated with the changes in protein structural characteristics, consistent with the findings of Sun, Huang, Li, Ang, Xu, and Huang (2019) [9]. Few studies have addressed protein changes with different meat processing treatments. The use of iTRAQ technology to determine the biochemical mechanisms and proteomic changes of sauce lamb tripe subjected to HPP and HT has not been reported.

In this study, iTRAQ technology was used to trace the proteomic changes of sauce lamb tripe after HPP and HT. Texture analysis was also performed to explore the changes in key proteins after HPP and HT, and the related biological mechanisms explaining the changes in the texture of sauce lamb tripe after different treatments.

## 2. Materials and Methods

### 2.1. Materials

The fresh lamb tripe was randomly purchased from Xinjiang Western Animal Husbandry Co., Ltd. (Shihezi, China). Urea and 4-(2-hydroxyethyl) piperazine-1-etha-nesulfonic acid (HEPES) was purchased from GibcoBRL (Shanghai, China). Ethylene diamine tetraacetic acid (EDTA) and Phenylmethanesulfonyl fluoride (PMSF) were purchased from Amrecso (Shanghai, China). Coomassie brilliant blue dye G250 was purchased from Amesco. Dithiothreitol (DTT) and iodoacetamide (IAM) were purchased from Promega (Beijing, China). Triethylammonium bicarbonate (TEAB), acrylamide and sodium dodecyl sulfate (SDS) and were purchased from Sigma-Aldrich (Shanghai, China). Acetonitrile was purchased from Fisher Scientific (Shanghai, China).

### 2.2. Preparation of Samples

Kazakh lambs (100–120 kg live weight, about ten months old and 24 h post-mortem) were randomly obtained from Xinjiang Western Animal Husbandry Co., Ltd. (Shihezi, China). Then, the fresh lamb tripe was put into fresh-keeping bags and quickly transported back to the laboratory to remove dirt and oil on the surface of the lamb tripe and then cleaned. The washed lamb tripe was sauced according to a certain recipe.

Sauce lamb tripe was vacuum-packed and divided into three groups (control group (CN), HPP, HT). Six samples (20 g per sample) were prepared for each group. HPP was performed at a pressure of 400 MPa, a pressure holding time of 15 min, and a temperature of 25 °C. HT was performed by putting samples in an 85 °C water bath for 40 min; when the core temperature of the samples measured by the probe was 85 °C, the samples were quickly transferred to ice water for cooling. The samples were stored at 4 °C. The protein samples were stored at −80 °C prior to analysis.

### 2.3. Determination of Texture Profile Analysis (TPA)

Samples were flattened and placed on a TA-XT plus texture analyzer (Stable Micro System Inc., Godalming, UK) for detection. The measurement conditions were: probe rate before measurement of 2.00 mm/s; probe rate at the time of measurement of 1.00 mm/s; probe rate after measurement of 2.00 mm/s; measured compression ratio of 50%; and a P5 probe model. Three replicates were determined for each treatment.

### 2.4. Protein Extraction

Next, 2 g of sauce lamb tripe sample was added to liquid nitrogen and ground, and then 3 mL of lysate buffer (8 M urea, 30 mM 4-(2-hydroxyethyl), piperazine-1-etha-nesulfonic acid (HEPES), 1 mM phenylmethanesulfonyl fluoride (PMSF), 2 mM ethylene diamine tetraacetic acid (EDTA), and 10 mM dithiothreitol (DTT)) were added, and the samples were homogenated for 15 min on ice. The resulting samples were loaded into 1.5 mL tubes and centrifuged at 4 °C and 20,000× *g* for 30 min. Next, the supernatants were mixed with four times the volume of precooled acetone, incubated at −20 °C for 3 h, and centrifuged at 20,000× *g* for 30 min at 4 °C. Lysis buffer was added to the precipitated material and the samples were treated by ultrasound for 5 min (pulse on, 2 s; pulse off, 3 s; power, 180 W). Next, DTT was added to a final concentration of 10 mM and the samples were incubated in a 56 °C water bath for 1 h. Iodoacetamide (IAM) was quickly added to a final concentration of 55 mM, and the samples were then incubated without agitation for 1 h in a dark room. After centrifugation at 4 °C and 20,000× *g* for 30 min, the supernatants were saved and assayed by the Bradford assay to quantify the protein concentration.

### 2.5. Protein Digestion and Labeling of Peptides

Samples of 100 μg proteins were transferred into 10 K ultrafiltration tubes and centrifuged at 4 °C and 14,000× *g* for 40 min. Next, 200 μL of 50 mM triethylammonium bicarbonate (TEAB) was used to resuspend the precipitated material and then subjected to centrifugation at 4 °C and 14,000× *g* for 40 min, after which the liquid was discarded. The above steps were repeated twice. Next, 11 μg/μL trypsin and 3.3 μg enzyme were added and the samples were incubated in a water bath at 37 °C for 24 h. The treated fluid was lyophilized, and then 30 μL TEAB (1:1 ratio of water: TEAB) was added to dissolve the peptides. Next, a mixture of isopropanol and a specific labeling reagent was added to the peptides, and samples were incubated in 25 °C for 2 h after mixing. Additionally, then, CN samples were labeled by 130N, 130C, and 131 isotope tags; HPP samples were labeled as 128N, 129N, and 129C isotope tags; and HT samples were labeled as 126, 127N, and 127C isotope tags.

### 2.6. Pre-Separation and Mass Spectrum Identification

Pre-separation was performed by high-performance liquid chromatography (HPLC) using a strong cation exchange column (Phenomenex, Torrance, CA, USA). High-pH reversed-phase liquid chromatographic separation of the pre-fractionated fractions was performed using an Acclaim PePmap C18 reverse-phase column (75 μm × 2 cm, 3 μm, 100 Ǻ Thermo Scientific, Waltham, MA, USA) mounted on a Dionex ultimate 3000 nano LC system (Dionex, Sunnyvale, CA, USA). Peptides were eluted using a gradient of 5–80% (*v*/*v*) acetonitrile in 0.1% formic acid over 45 min at a flow rate of 300 nL min^−1^. The eluates directly entered Q-Exactive MS (Thermo Fisher Scientific, Waltham, MA, USA) in positive ion mode with a full MS scan from 350–2000 *m*/*z*, full scan resolution at 70,000, and MS/MS scan resolution at 17,500. The MS/MS scan was performed with minimum signal threshold 10^5^ and isolation width of 2 Da. To evaluate the iTRAQ-labeled samples, two MS/MS acquisition modes with higher collision energy dissociation (HCD) were employed. To optimize the MS/MS acquisition efficiency of HCD, the normalized collision energy (NCE) was systemically examined at 28, with 20% steps.

### 2.7. Western Blot

The protein samples were subjected to SDS-PAGE electrophoresis using 12% polyacrylamide separating gel. After the gel was transferred to PVDF membrane, blotting was performed using the procedure of Laemmli (1970) [10]. The PVDF membrane was incubated in 5% skim milk with shaking for 1 h at room temperature. Samples were incubated at 4 °C overnight in a solution of primary antibody (anti-CNN1 or anti-ACTB from Bioss Biotechnology Inc. (Bioss, Beijing, China)) diluted with 5% skim milk in 0.5% TBST (1:1000). The PVDF membrane was washed with TBST three times, and each wash lasted for 5 min. The secondary antibody was diluted in TBST (1:10,000) and incubated with the membrane for 30 min, followed by three washes of 5 min with TBST at room temperature. The PVDF membrane was then incubated in a mixture of ECLA and ECLB reagents and imaged using a ChemiScope Capture imaging system (Clinx, Shanghai, China). The included ChemiScope analysis software was used for grayscale analysis.

### 2.8. Bioinformatics and Statistical Analysis

The significant differences of the mass spectrometry results was assessed by one-way analysis of variance (ANOVA). Proteins with *p*-value less than 0.05, ratio ≥ 1.2, or ratio ≤ 0.83 were selected as DEPs. Gene ontology (GO) function annotation (cellular function, biological function, and molecular function) was conducted for the DEPs (http://www.geneontology.org/ (accessed on 20 November 2018)). Protein–protein interaction was analyzed (http://string-db.org/ (accessed on 20 November 2018)). SPSS 24.0 was used to analyze the TPA of different groups and the results are expressed as means ± standard deviations (SD). Differences were considered significant at *p* < 0.05. The experiment was repeated three times.

## 3. Results

### 3.1. TPA Analysis

Texture is an important indicator of the tenderness of meat, and the texture change in meat after HPP is closely related to the changes in proteins. The texture was measured for the three kinds of samples and the results are shown in Table 1. The hardness and gumminess in the HT group were significantly different from those in the CN group (*p* < 0.05). Additionally, the springiness and chewiness in the HPP group were significantly different from those of the CN group (*p* < 0.05), and the parameters in the HPP and HT groups were significantly different (*p* < 0.05). Compared to HT and CN, HPP increased the springiness and chewiness, and improved the quality and taste of the lamb tripe to a certain extent. The HT of meat can cause changes in the degradation of myofibrils and connective tissue [11]. Furthermore, changes in the texture of meat products are also observed due to HPP. Yamira, Mauricio, Anja, Janssen, Gipsy, and Mario (2020) [12] showed that high pressure changed the secondary structure of proteins and improved protein digestibility.

### 3.2. Proteomic Analysis

Figure 1 shows a volcano map of the identified proteins, where red indicates up-regulated differentially expressed proteins (DEPs) and green indicates down-regulated DEPs. The number of DEPs after HPP was significantly higher than that after HT, indicating a more obvious effect of HT on the proteins of sauce lamb tripe. Figure 2 shows Venn diagrams of the DEPs in the three groups. A total of 484 DEPs were identified in CN/HPP comparison, with 283 DEPs up-regulated and 202 DEPs down-regulated; 398 DEPs were identified in the CN/HT comparison, with 243 DEPs up-regulated and 155 DEPs down-regulated. A total of 289 DEPs were identified in the HPP/HT group, including 159 up-regulated DEPs and 130 down-regulated DEPs. The intersection of the CN/HT and HPP/HT comparisons was defined as Group A (43 DPEs), the intersection of the CN/HPP and HPP/HT comparisons was defined as Group B (92 DEPs), and the intersection of the CN/HT, CN/HPP, and HPP/HT comparisons was defined as Group C (43 DEPs). The differences in these comparisons indicated that high pressure and HT had different effects on the proteins in sauce lamb tripe.

According to the observed differences in texture, proteins with similar difference trends were identified, including many metabolic enzymes, structural proteins, and regulatory proteins. These are shown in Table A1 (uncharacterized proteins are not shown). These included 22 DEPs in Group A, 58 DEPs in Group B, and 27 DEPs in Group C.

### 3.3. Western Blot Analysis

Western blot analysis can be used to verify the results of protein expression. Calponin-1 (CNN1) and actin-cytoplasmic 1 (ACTB) existed in Group A and Group B, respectively, and they both had higher protein scores (CNN1, 37,163.94; ACTB, 31,624.72) and more significant difference multiples (CNN1, 1.637-fold; ACTB, 1.567-fold) (Table A1). Therefore, using these two proteins for verification helped determine the accuracy of DEPs after ultrahigh pressure and HT. As shown in Figure 3, after HT, the signal corresponding to CNN1 appeared lighter than its intensity in the samples from the other two groups, indicating that HT caused a decrease in the expression level of CNN1. As shown in Figure 4, gray value analysis revealed that the expression level of CNN1 was significantly decreased (1.637-fold) after HT (*p* < 0.01), and the expression level of ACTB after HPP was also significantly decreased (*p* < 0.05). After high-pressure processing, the expression level of ACTB was decreased (1.567-fold) (Table A1). Overall, these results were consistent with the observed changes in protein expression by proteomic determination.

### 3.4. Gene Ontology Nalysis of DEPs

Gene ontology (GO) is a method for the functional annotation of proteins, with functions divided into cellular components, biological processes, and molecular functions. The results of the GO analysis are shown in Figure 5. The DEPs in Group A mainly included cellular components, the DEPs in Group B mainly included cellular components and biological processes, and the DEPs in Group C mainly included cellular composition and molecular function. HPP had a significant impact on the proteins involved in cellular components and biological processes. The molecular functions of the DEPs after HPP were mainly molecular function (GO: 0003674), binding (GO: 0005488), transporter activity (GO: 0005215), and catalytic activity (GO: 0003824), and those of the DEPs after HT were mainly binding (GO: 0005488), catalytic activity (GO: 0003824), transporter activity (GO: 0005215), and kinase activity (GO: 0016301). Therefore, the changes in the functional proteins and kinase activity might partially explain the different textures of sauce lamb tripe after different treatments. After HT, keratin 1 (KRT1, 1.470-fold) appeared to have structural molecular activity; this did not appear after HPP. These DEPs with molecular activity might explain the texture differences in the meat samples that received the different treatments.

### 3.5. Protein–Protein Interaction Networks of DEPs

The results of protein–protein interaction (PPI) network analysis of the DEPs listed in Table A1 are shown in Figure 6. The PPI included 10 DEPs from Group A (ACTA2, CNN1, MDH2, FN1, ETFA, GOT1, COX4I1, ME1, EFEMP1, and AEBP1), 31 from Group B (FLNA, ACTB, TPM4, COL1A1, FLNC, COL1A2, MYH10, APOA1, FLNB, LOC101106313, YWHAG, MYH8, COL3A1, COL4A2, AHSG, MB, TNNT3, LOC101112249, APOA4, PVALB, LOC101109421, TTR, TTN, ENSOARG00000009612, ENSOARG00000006272, FETUB, HRG, ARGC3, ARGC3, and HDDC2), and 7 from Group C (TPM3, COL5A2, MYL2, COL5A1, FKBP3, TNNI2, and LOC101112491). The DEPs in Group A were mainly composed of structural proteins, enzymes, and protein subunits, the DEPs in Group B were mainly composed of structural proteins and protein subunits, and the DEPs in Group C were mainly structural proteins (Figure 6). These structural proteins were mainly involved in muscle contraction [13]. The main proteins involved in the protein–protein interactions were structural proteins, and hence were of particular interest.

The structural proteins of group A included actin alpha 2 (ACTA2), calponin-1 (CNN1), beta-actin, myosin light chain 3 (MYL3), myosin heavy chain 7 (MYH7), fibronectin 1 (FN1), and keratins (KRT1 and KRT8). Group B included actin-cytoplasmic 1 (ACTB), tropomyosin (TPM1 and TPM4), collagen isoforms (COL1A1, COL1A2, COL3A1, COL4A2 and COL5A3), filamin (FLNA, FLNB and FLNC), myosin heavy chain (MYH8 and MYH10), 14-3-3 protein γ isoform (YWHAG), parvalbumin (PVALB), titin (TTN), and actin-related protein 2/3 complex subunit 3 (ARPC3). The structural proteins in Group C included tropomyosin 3 (TMP3), collagen isoforms (COL6A1, COL5A2 and COL5A1), myosin light chain 1 transcription variant 2 (MYL1b), calmodulin (CALM2), keratin 77 (KRT77), fibrinogen β chain (FGB), and myosin light chain 2 (MYL2). According to the composition of the DEPs in the three groups, CNN1, FN1, and keratin (KRT) in Group A, and the collagen isoform, FLN, YWHAG, TTN, and ARPC3 in Group B might be the key DEPs that cause the change in texture after the different treatments of sauce lamb tripe.

### 3.6. Potential Key Proteins Associated with Texture

CNN1 is a regulator of smooth muscle contraction and responsiveness to contraction activation; it can regulate contractile actin-myosin filaments and the non-contractile actin cytoskeleton of smooth muscle cells [14]. It participates in the formation of actin structure and tissues. Its expression level was significantly increased after HT. An interaction between CNN1, actin, and myosin was also shown in the PPI analysis (Figure 6). The interaction of these proteins might be one of the ways that CNN1 regulates texture. Keratin had structural molecular activity, and its cellular composition, biological processes, and molecular function annotations indicated its involvement in the formation of intermediate filaments of smooth muscle cells (Figure 5). Differences were found in the levels of KRT8 and KRT10, but previous proteomic analyses of meat product quality identified that the differential expression of KRT did not correlate with the quality of the meat products [15,16]. FN1 was related to the formation and transformation of muscle fat, and intramuscular fat content had a strong relationship with the tenderness of meat products [17]. The expression level of FN1 was significantly increased after HT (Table A1), causing a decrease in the fat content in smooth muscle and the hardness of sauce lamb tripe.

Three main types of FLN were identified (FLNA, FLNB, and FLNC) (Table A1). FLNC is mainly expressed in skeletal muscle and the myocardium [18]. FLNA is a large multi-domain homodimeric actin-binding protein that promotes the mechanical stability of cells, enhances the mechanical protection of cells exposed to external physical forces, and interacts with a variety of proteins to regulate cell adhesion [19]. FLNA expression was significantly up-regulated (0.779-fold) after HPP, indicating that this treatment improved the stability and the adhesion of cells, altering their susceptibility to external forces. FLN had a certain effect on meat tenderness, but was not fully characterized [20]. Collagen formed an ordered hierarchy with the fibrils (types I, II, and III) and networks (type IV), providing elasticity, stability, and support to the tissue [21,22]. Five collagen isoforms (COL1A1, COL1A2, COL3A1, COL4A2, and COL5A3) significantly decreased after high pressure (Table A1). According to Brigitte and Mohamed (2020) [13], in terms of the classification of the biomarkers of beef tenderness, COL4A1 was related to contraction and the related proteins. At the same time, the sauce lamb tripe had higher springiness and chewiness (Table 1). Chanporn, Ronachai, Panneepa, Apichaya, and Kazeem (2020) [23] found that goat meat with a higher total collagen value was less tender. Therefore, the effects of collagen isoform expression on muscle might explain the increase in springiness and chewiness after HPP. TTN is one of three specific families of structural motifs in the main structure of actin, which acts as a molecular spring [24]. A report on smooth muscle myosin and its role in the organization of myosin assembly described TTN as an elastic cytoskeletal molecule that is widely present in muscle and non-muscle cells [25]. The expression levels of TTN decreased significantly after HPP (Table A1). Lana and Zolla (2016) [20] reported that TTN had a structural function and affected the tenderness of the meat, but different muscle types had different degradation patterns affecting tenderness. In addition to TNN, 14-3-3 proteins also showed significantly different expressions. The 14-3-3 proteins were highly conserved, and many organisms expressed multiple isoforms of the protein. Seven 14-3-3 protein isoforms (β, ɛ, η, γ, τ, ζ, and σ) have been identified in mammals; these proteins are involved in the regulation and coordination of many cellular processes, including apoptosis, metabolism, the transcriptional regulation of gene expression, and DNA damage [26,27]. The proteomic analysis of Maremmana beef identified that YWHAG protein up-regulation related to increased tenderness [28]. In this study, YWHAG expression showed a significant decrease (1.325-fold) after HPP, but demonstrated increased springiness and chewiness. ARPC3 was significantly up-regulated after HPP (Table A1), and this protein could regulate the formation of the actin filament network [29]. Poleti, Regitano, and Souza (2018) [30] reported that the intramuscular fat of beef impacted tenderness, and observed a decreased expression of TTN and ARPC2 in beef with high intramuscular fat deposition.

Compared with HT, the sauce lamb tripe subjected to HPP contained decreased levels of collagen subunits, TPM3, myosin light chain, and calmodulin. This sauce lamb tripe exhibited higher springiness and chewiness (Table 1). Tropomyosin and myosin light chain are important structural proteins that affect muscle tenderness. Both can interact with actin, thereby affecting the texture of smooth muscle. Overall, the changed structural proteins in the HT group were mainly involved in the muscle contraction of smooth muscle, or improved the adhesion of muscle cells to improve the hardness and gumminess of the sauce lamb tripe. The structural proteins in the HPP group affected springiness. These proteins increased the springiness and chewiness of the smooth muscle of the sauce lamb tripe by participating in the formation of actin or binding with actin.

## 4. Conclusions

The differential proteomics of sauce lamb tripe with different secondary pasteurization were analyzed. The results showed that the changes in structural proteins affected the texture of sauced lamb tripe after HPP or HT. CNN, FN1 and KRT showed differences after HT. Moreover, TMP, FLN, YWHAG, TTN, collagen isoforms, and ARPC3 exhibited changes after HPP. The changes in the expression levels of these proteins affected the texture of the sauce lamb tripe. Further studies should investigate their specific mechanisms of action.

## Figures and Tables

**Figure 1 foods-11-01166-f001:**
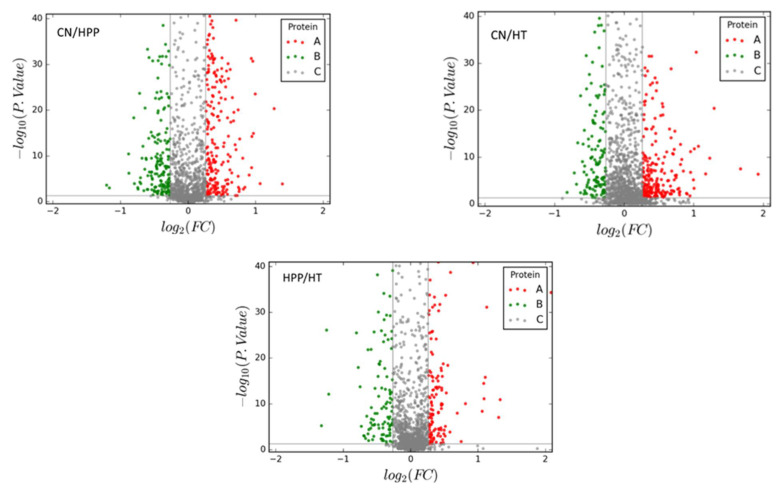
Volcano map of HPP and HT. The red dots in the volcano map represent the up-regulated protein, the green dots represent the down-regulated proteins, and the gray dots represent proteins that were not differential.

**Figure 2 foods-11-01166-f002:**
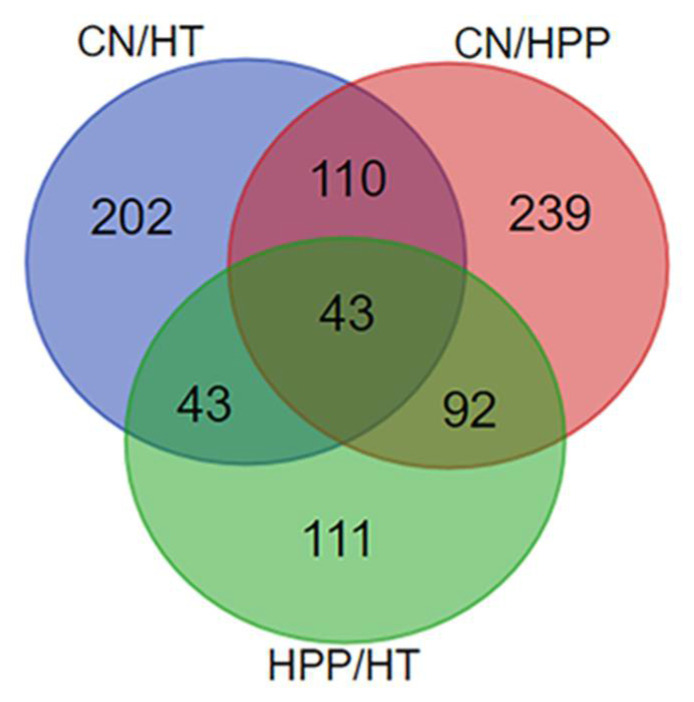
Venn diagram of DEPs. The quantities of differential proteins after treatment are clearly shown in Figure 2.

**Figure 3 foods-11-01166-f003:**
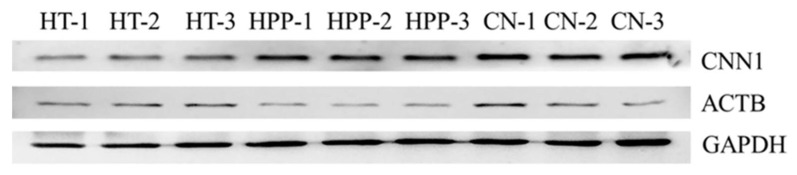
Western blots of CNN1, ACTB and GAPDH. CNN1 and ACTB in 3 come from Group A and Group B, respectively. GAPDH is the reference protein. CN is the sample without treatment, HT is the sample after heat treatment, and HPP is the sample after high-pressure treatment. 1, 2 and 3 are three parallels of the same sample.

**Figure 4 foods-11-01166-f004:**
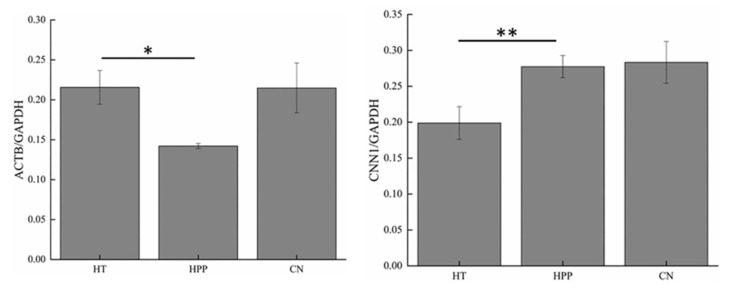
Analyses of Western blot bands. The band obtained by WB was analyzed by gray value, and the result was expressed as the gray value of the target band/the gray value of the reference protein. “*” stands for significant difference *p* < 0.05; “**” stands for significant difference *p* < 0.01.

**Figure 5 foods-11-01166-f005:**
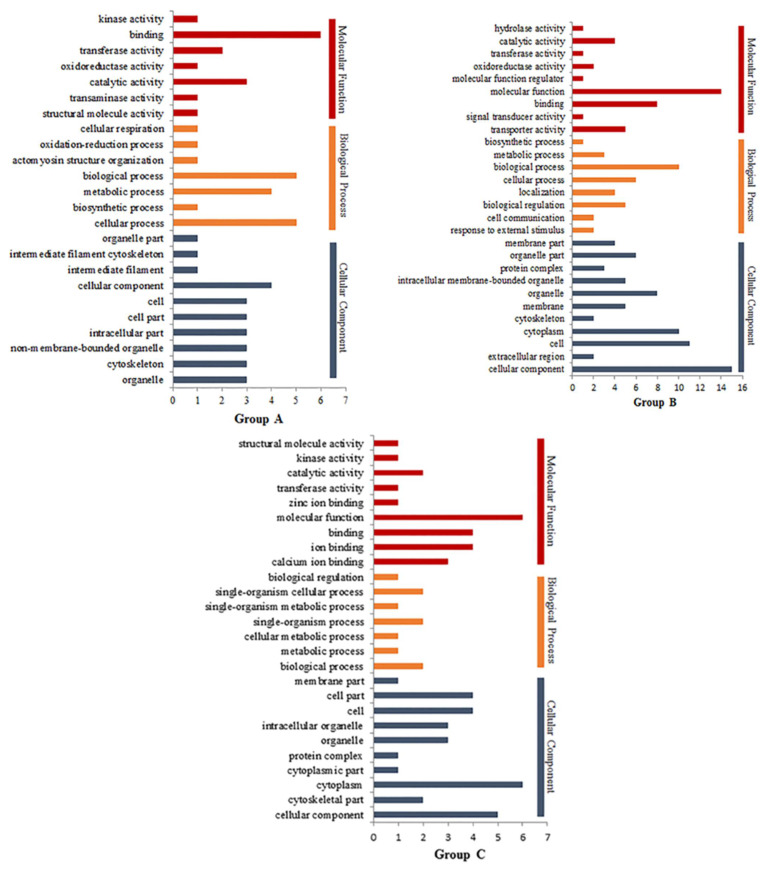
GO analyses of three groups. The figure only shows the cellular components, biological processes and molecular functions of the three groups (**A**–**C**).

**Figure 6 foods-11-01166-f006:**
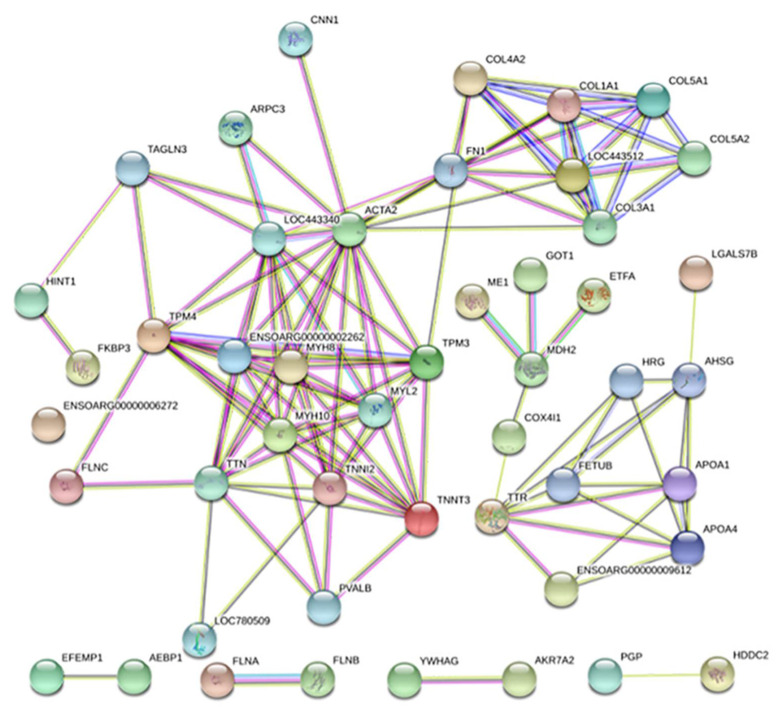
Protein–protein interaction networks of DEPs. Empty nodes—proteins of unknown 3D structure; filled nodes—some 3D structure is known or predicted. The interaction includes three parts (known interactions (
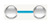
 from curated databases; 
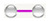
 experimentally determined), predicted interactions (
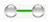
 gene neighborhood; 
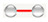
 gene fusions; 
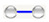
 gene co-occurrence) and others (
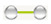
 textmining; 
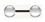
 co-expression; 
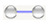
 protein homology)). Some proteins without protein identifiers do not appear in the figure.

**Table 1 foods-11-01166-t001:** Texture results of three samples.

	CN	HT	HPP
Hardness	247.63 ± 50.366 ^a^	435.314 ± 87.039 ^b^	255.633 ± 64.275 ^a^
Springiness	0.889 ± 0.054 ^a^	0.912 ± 0.125 ^a^	2.401 ± 0.2 ^b^
Gumminess	206.986 ± 35.032 ^a^	369.368 ± 51.983 ^b^	240.797 ± 58.856 ^a^
Chewiness	185.185 ± 40.766 ^a^	332.062 ± 4.675 ^a^	584.52 ± 180.439 ^b^

Note: The results in the table are Mean ± SD; the lowercase letters are the results of the difference significance analysis in the horizontal rows (*p* < 0.05).

## Data Availability

No new data were created or analyzed in this study. Data sharing is not applicable to this article.

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
