# Peer review of "Application of iTRAQ Technology to Identify Differentially Expressed Proteins of Sauce Lamb Tripe with Different Secondary Pasteurization Treatments"

_foods, 2022, doi:10.3390/foods11081166_

Round 1

Reviewer 1 Report

Comments:

The manuscript is interesting and presents novel information on textural changes in sauce lam tripe treated with secondary pasteurization (high pressure processing and heat treatment). Differential proteomics allowed to identify changes in the expression of structural proteins related to texture. Minor details in the methodology must be considered in order to ensure the experiment reproducibility.

In addition, it is necessary to justify the practical relevance of the study, i.e. the advantages of applying high pressure and/or heat treatment in improving the quality of lamb tripe. This could extrapolate the results for potential applications in similar products such as chicken and beef tripes, which are widely consumed in other countries.

Finally, a sensory study related to texture parameters would allow evaluating whether high-pressure processing and/or heat treatment really improves the attributes that are relevant to consumers. A change in the texture of chicken tripes does not guarantee consumer acceptance or preference.

Some specific issues are the following:

Line 66. How much fresh lamb tripe were used in the study? Were all tripes from the same animal? Are the characteristics of the lambs known (live weight, feeding, age, etc.) How do you ensure the homogeneity of the experimental units for the allocation of treatments?

Line 78. The idea is not clear. The lamb tripes were sauced according to a certain recipe? please describe it.

Line 79. Describe the characteristics of the packaging (trademark, bag thickness or caliber, oxygen permeability, etc.).

Line 80. How much sample did each vacuum pack contain? How many vacuum packed samples were used? Was it 3? One per group? It's not clear. Why have you used 20 g samples? were the samples vacuum packed during the treatment?

Line 83. Did you measure the temperature of the samples or only the temperature of the bath? Have you used a probe to measure the temperature?

Line 167. Further discussion of textural results regarding the effect of high pressure and heat treatment on quality of lamb tripes is needed. For example, the increase in hardness when high pressures are applied seems to be a negative effect. What about the other texture properties? the fact that the gumminess increases is a positive or negative effect?

Author Response

We appreciate the thorough evaluation and affirmation of our study by the reviewer. In the revised version of our manuscript, we address essentially all of the points raised by the reviewers, and we think the resulting study is much improved. We have made corresponding modifications, which have been marked in red in the text. Specific revisions and responses to each comment are provided in detail below:

  1. Line 66. How much fresh lamb tripe were used in the study? Were all tripes from the same animal? Are the characteristics of the lambs known (live weight, feeding, age, etc.) How do you ensure the homogeneity of the experimental units for the allocation of treatments?

ResponseThanks for your suggestions. All tripes come from the same animal (Kazakh lamb). In the pretreatment of the experimental raw materials, 5 lamb tripes (with dirt and grease from the surface of the lamb tripe removed) were cut together, and 18 samples of similar quality and size were obtained (20±0.5g). The lamb tripes were randomly divided into 3 groups (CN, HT, HPP), there were 6 samples in each group for index determination. The relevant content in the manuscript has been re-edited: The Kazakh lamb (100-120 kg live weight, about ten months old and 24 h post-mortem) were randomly obtained from Xinjiang Western Animal Husbandry Co., Ltd. (Shihezi, China). (Line 77-79)

  1. Line 78. The idea is not clear. The lamb tripes were sauced according to a certain recipe? please describe it.

Response: Thanks for your suggestions. In the process of experiment, marinate the freshly washed lamb tripe. Condiment dosage: 200 g lamb tripe, 2.0 g salt, 12 g cooking wine, 0.4 g star anise, 2 g cinnamon, 0.06 g cumin, 0.26 g peppercorn, 1.58 g chicken essence, 7 g soy sauce, 0.9 g dried chili, 300 mL water , the halogenation time was 60 min.

  1. Line 79. Describe the characteristics of the packaging (trademark, bag thickness or caliber, oxygen permeability,).

Response: Thanks for your suggestions. Packaging material is polyvinylidene chloride/polyethylene (PVDC/PE). Bag thickness: 0.2 mm; oxygen permeability < 10.65 cm3/(m2·24h); Water vapor permeability < 2.671 cm3/(m2·24h).

  1. Line 80. How much sample did each vacuum pack contain? How many vacuum packed samples were used? Was it 3? One per group? It's not clear. Why have you used 20 g samples? were the samples vacuum packed during the treatment?

Response: Thanks for your suggestions. In the preliminary experiment, the marinated lamb tripe was vacuum-packed in bags (20g / bag), a total of 18 bags. Six bags (20 g / bag) were prepared for each group for index determination. 20g of sample was used for texture determination and proteomics study, and meet the requirements of experimental dosage. The samples were kept in a vacuum state during the during the treatment.

  1. Line 83. Did you measure the temperature of the samples or only the temperature of the bath? Have you used a probe to measure the temperature?

Response: Thanks for your suggestions. Relevant suggestions have been supplemented in the manuscript. In the experiment, we measured the core temperature of the sample through the probe and confirmed that it reached the required temperature. (Line85-86)

  1. Line 167. Further discussion of textural results regarding the effect of high pressure and heat treatment on quality of lamb tripes is needed. For example, the increase in hardness when high pressures are applied seems to be a negative effect. What about the other texture properties? the fact that the gumminess increases is a positive or negative effect?

Response: Thanks for your suggestions. The relevant content in the manuscript has been re-edited. Compared to HT and CN, HPP increased the springiness and chewiness, and improved the quality and taste of lamb tripes to a certain extent.(Line 167-168)

Gumminess likely reflect changes in water distribution, solubilization of proteins and the formation of modified protein products. For meat products, high gumminess is detrimental to the taste of meat products (Zhu et al.; 2022; Schreuders et al.; 2021 )1, 2.

  1. Zhu, Y.; Yan, Y.;  Yu, Z.;  Wu, T.; Bennett, L. E., Effects of high pressure processing on microbial, textural and sensory properties of low-salt emulsified beef sausage. Food Control 2022, 133.
  2. Schreuders, F.; Schlangen, M.;  Kyriakopoulou, K.;  Boom, R. M.; Goot, A. J. F. C., Texture methods for evaluating meat and meat analogue structures: A review. 2021,  (1), 108103.

Reviewer 2 Report

Dear authors,

firstly, please excuse the late reply. 

Your work describes an interesting and important work on proteomics of the food, which is not always properly covered in scientific work

Some general remarks:

  1. Figure captions: Please describe the results shown and not describe the method used. For example in Figure 2 you wrote: Figure 2. Venn diagram of DEPs. The Venn diagram can be used to study and express the "set prob-195 lem" in middle mathematics, including "intersection", "union", and "complement" and so on. The 196 quantities of differential proteins after treatment are clearly shown in the Figure 2. The reader does not need the description of the Ven Diagramm, please change the captions accordingly.
  2. Please have a native speaker check the English language and style.
  3.  You write that "Desalted peptide 120 mixtures were loaded onto an Acclaim PePmap C18-reverse phase column (75 μm×2 cm, 121 3 μm, 100 Ǻ Thermo Scientific) and mounted on a Dionex ultimate 3000 nano LC sys-122 tem(Dionex, Sunnyvale, USA)". I assume that you have forgotten to mention the separation column, or did you really achieve the results described on a trap column?
  4. For proteomics experiments, the raw data shall be uploaded to some of the data repositories, eg.g., PRIDE and made available to reviewers and peers. 

Otherwise, I have no substantial comments.

Kind regards,

Author Response

We appreciate the thorough evaluation and affirmation of our study by the reviewer. In the revised version of our manuscript, we address essentially all of the points raised by the reviewers, and we think the resulting study is much improved. We have made corresponding modifications, which have been marked in red in the text. Specific revisions and responses to each comment are provided in detail below:

  1. Figure captions: Please describe the results shown and not describe the method used. For example in Figure 2 you wrote: Figure 2. Venn diagram of DEPs. The Venn diagram can be used to study and express the "set prob-195 lem" in middle mathematics, including "intersection", "union", and "complement" and so on. The 196 quantities of differential proteins after treatment are clearly shown in the Figure 2. The reader does not need the description of the Ven Diagramm, please change the captions accordingly.

Response: Thanks for your very considerate input. Superfluous notes in the figure captions have been removed from the manuscript.

  1. Please have a native speaker check the English language and style.

Response: Thanks for your advice. After the revision, we followed your advice to find a native speaker of English (preferably one versed in the topic) read the final version of the paper for flow and style of the language. Finally, a revised final version has been submitted. The revised content have been highlighted in red.

  1. You write that "Desalted peptide 120 mixtures were loaded onto an Acclaim PePmap C18-reverse phase column (75 μm 2 cm, 121 3 μm, 100 Ǻ Thermo Scientific) and mounted on a Dionex ultimate 3000 nano LC sys-122 tem(Dionex, Sunnyvale, USA)". I assume that you have forgotten to mention the separation column, or did you really achieve the results described on a trap column

Response: Thanks for your very considerate input. The relevant content in the manuscript has been re-edited.(Line 124-126)

  1. For proteomics experiments, the raw data shall be uploaded to some of the data repositories, eg.g. , PRIDE and made available to reviewers and peers.

Response: Thanks for your suggestions. The raw data have been uploaded to reviewers.
